First insights into the diversity of gill monogeneans of ‘Gnathochromis’ and Limnochromis (Teleostei, Cichlidae) in Burundi: do the parasites mirror host ecology and phylogenetic history?

Kmentová Nikol 1 kmentovan@mail.muni.cz
Gelnar Milan 1
Koblmüller Stephan 2 3
Vanhove Maarten P.M. 1 4 5 6
1 Department of Botany and Zoology, Masaryk University , Brno , Czech Republic
2 Institute of Zoology, University of Graz , Graz , Austria
3 Institute of Vertebrate Biology, Academy of Sciences of the Czech Republic , Brno , Czech Republic
4 Biology Department, Royal Museum for Central Africa , Tervuren , Belgium
5 Laboratory of Biodiversity and Evolutionary Genomics, Deparment of Biology, University of Leuven , Leuven , Belgium
6 Capacities for Biodiversity and Sustainable Development, Royal Belgian Institute of Natural Sciences , Brussels , Belgium
Esteban María Ángeles
Electronic publication date: 2016 Jan 25
Publication date: 2016
Volume: 4
Electronic Location ID: e1629
Received 2015 Nov 2; Accepted 2016 Jan 5
Copyright: ©2016 Kmentová et al.
Copyright year: 2016
Copyright holder: Kmentová et al.
License: This is an open access article distributed under the terms of the Creative Commons Attribution License, which permits unrestricted use, distribution, reproduction and adaptation in any medium and for any purpose provided that it is properly attributed. For attribution, the original author(s), title, publication source (PeerJ) and either DOI or URL of the article must be cited.
License URL: https://creativecommons.org/licenses/by/4.0/

Keywords: Cichlidogyrus, Lake Tanganyika, Ectoparasites, Limnochromini, Tropheini

Funding: Czech Science Foundation GBP505/12/G112-ECIP This study was financially supported by the Czech Science Foundation (GBP505/12/G112-ECIP). The funders had no role in study design, data collection and analysis, decision to publish, or preparation of the manuscript.

==============================
Monogenea is one of the most species-rich groups of parasitic flatworms worldwide, with many species described only recently, which is particularly true for African monogeneans. For example, Cichlidogyrus, a genus mostly occurring on African cichlids, comprises more than 100 nominal species. Twenty-two of these have been described from Lake Tanganyika, a famous biodiversity hotspot in which many vertebrate and invertebrate taxa, including monogeneans, underwent unique and spectacular radiations. Given their often high degrees of host specificity, parasitic monogeneans were also used as a potential tool to uncover host species relationships. This study presents the first investigation of the monogenean fauna occurring on the gills of endemic ‘Gnathochromis’ species along the Burundese coastline of Lake Tanganyika. We test whether their monogenean fauna reflects the different phylogenetic position and ecological niche of ‘Gnathochromis’ pfefferi and Gnathochromis permaxillaris. Worms collected from specimens of Limnochromis auritus, a cichlid belonging to the same cichlid tribe as G. permaxillaris, were used for comparison. Morphological as well as genetic characterisation was used for parasite identification. In total, all 73 Cichlidogyrus individuals collected from ‘G.’ pfefferi were identified as C. irenae. This is the only representative of Cichlidogyrus previously described from ‘G.’ pfefferi, its type host. Gnathochromis permaxillaris is infected by a species of Cichlidogyrus morphologically very similar to C. gillardinae. The monogenean species collected from L. auritus is considered as new for science, but sample size was insufficient for a formal description. Our results confirm previous suggestions that ‘G.’ pfefferi as a good disperser is infected by a single monogenean species across the entire Lake Tanganyika. Although G. permaxillaris and L. auritus are placed in the same tribe, Cichlidogyrus sp. occurring on G. permaxillaris is morphologically more similar to C. irenae from ‘G.’ pfefferi, than to the Cichlidogyrus species found on L. auritus. Various evolutionary processes, such as host-switching or duplication events, might underlie the pattern observed in this particular parasite-host system. Additional samples for the Cichlidogyrus species occuring on G. permaxillaris and L. auritus are needed to unravel their evolutionary history by means of (co-)phylogenetic analyses.

Introduction

Cichlid fishes (Cichlidae) are considered an ideal study system for evolutionary biologists because of their remarkable species richness, high rates of speciation and often high levels of endemicity, derived from diverse speciation and adaptive radiation processes (Salzburger et al., 2005; Turner, 2007; Muschick, Indermaur & Salzburger, 2012). Studies about cichlid adaptation mechanisms provided important information, generally applicable in evolutionary biology (Kocher, 2004; Koblmüller, Sefc & Sturmbauer, 2008). Cichlids range from Central and South America, across Africa, Iran, the Middle East and Madagascar to India and Sri Lanka, but most species are concentrated in the Neotropics and in Africa (Chakrabarty, 2004). A place famous for its extraordinary cichlid diversity is Lake Tanganyika in East Africa (Koblmüller, Sefc & Sturmbauer, 2008). It is considered a prime study area for evolutionary research as its cichlids show the greatest diversity in speciation mechanisms of all the African Great Lakes’ cichlid fishes (Salzburger et al., 2002; Salzburger, 2009). In Lake Tanganyika, there are more than 200 described cichlid species belonging to 53 genera (Snoeks, 2000; Takahashi, 2003; Koblmüller, Sefc & Sturmbauer, 2008), usually classified into 15 tribes (Takahashi, 2003; Takahashi, 2014).

Although cichlids have been subjects of interest for many decades, there are still gaps in the understanding of their phylogenetic history and taxonomy (Koblmüller, Sefc & Sturmbauer, 2008). According to recent molecular findings, the two species of ‘Gnathochromis’, G. permaxillaris (LR David, 1936) and ‘G.’ pfefferi (GA Boulenger, 1898) belong to different cichlid tribes (Limnochromini and Tropheini, respectively) and their classification therefore needs revision (Salzburger et al., 2002; Duftner, Koblmüller & Sturmbauer, 2005; Koblmüller et al., 2010; Muschick, Indermaur & Salzburger, 2012; Kirchberger et al., 2014). A possible source for a better understanding of cichlid taxonomy and phylogeny, and a particularly diverse group of organisms in Lake Tanganyika, are monogenean parasites (Mendlová et al., 2012; Vanhove et al., 2015; Van Steenberge et al., 2015). Monogenea P-J Van Beneden, 1858 is a group of parasitic flatworms mainly occurring on fish gills, skin and fins (Pugachev et al., 2009). These often tiny animals have a direct life cycle, and relatively strong host specificity was reported on cichlid hosts (Pariselle & Euzet, 2009; Gillardin et al., 2012; Muterezi Bukinga et al., 2012; Řehulková, Mendlová & Šimková, 2013), which makes them an ideal model for investigating co-evolutionary processes in host-parasite systems (Pouyaud et al., 2006). While there is no published data available for the monogenean fauna on any of the tribe members of Limochromini, there is a pretty good record regarding the Cichlidogyrus diversity on the various species within Tropheini, with a high degree of host specificity and phylogenetic congruence (Vanhove et al., 2015). ‘Gnathochromis’ pfefferi, Limnotilapia dardennii (GA Boulenger, 1899) and ‘Ctenochromis’ horei (A Günther, 1894) are infected by a single dactylogyridean monogenean species each: Cichlidogyrus irenae, C. steenbergei and C. gistelincki C Gillardin, MPM. Vanhove, A Pariselle et al., 2012, respectively (Gillardin et al., 2012). Astatotilapia burtoni (A Günther 1894), a haplochromine cichlid closely related to the Tropheini (Koblmüller et al., 2008; Meyer, Matschiner & Salzburger, 2015), is infected by C. gillardinae F Muterezi Bukinga, MPM Vanhove, M Van Steenberge et al., 2012 (Muterezi Bukinga et al., 2012). These observations are hitherto only based on reports from several localities along the Congolese, Tanzanian and Zambian coasts of the lake (Gillardin et al., 2012; Muterezi Bukinga et al., 2012; Vanhove et al., 2015). Thorough sampling covering as many host localities as possible is, however, needed to conclude about the full extent of a species’ parasite fauna (Price & Clancy, 1983; Brooks et al., 2006; Caro, Combes & Euzet, 1997).

As mentioned above, ‘Gnathochromis’ is a polyphyletic genus and no comparison of the parasite fauna of its two species has been performed to date. Do the parasites reflect the phylogenetic position and ecological characteristics of their hosts? We investigated the monogenean fauna of both ‘Gnathochromis’ species to answer the following questions:

(1) Does the Burundese population of ‘G.’ pfefferi confirm that this host is only infected by a single species of Cichlidogyrus?

(2) Since ‘Gnathochromis’ is considered polyphyletic, is the phylogenetic distinctness of its two representatives also reflected in their parasite fauna?

Material & Methods

Sampling

Fish specimens were obtained from commercial fishermen along the Burundese coastline of Lake Tanganyika. Two ‘G.’ pfefferi individuals from Mvugo (4°15′S, 29°34′E) and four from Mukuruka (4′14′S, 29°33′E) were examined, as well as seven G. permaxillaris and six Limnochromis auritus (GA Boulenger, 1901) individuals from Bujumbura (3°23′S 29°22′E) (Fig. 1). Maps were created using SimpleMappr software (Shorthouse, 2010). The latter species was included to allow a comparison between the monogeneans of G. permaxillaris and another member of the Limnochromini, a tribe from which no monogeneans have been described previously. Fish were sacrificed by severing the spinal cord and dissected immediately. Gills were removed according to the standard protocol of Ergens & Lom (1970) and immediately preserved in pure ethanol in plastic tubes until further inspection in the lab. Some fresh gills were also inspected in situ for monogenean parasites using dissecting needles and a stereomicroscope. Slides prepared in situ were fixed in glycerine ammonium picrate (GAP) (Malmberg, 1957) or in Hoyer’s solution (Humason, 1979). Monogeneans were isolated in the lab using a dissecting needle and an Olympus SZX7 stereomicroscope. They were mounted on a slide under a cover slip. Parasite individuals used for genetic characterisation were identified using an Olympus BX51 microscope with incorporated phase contrast at a magnification of 100× (oil immersion, 10× ocular) with Micro Image software and photographed for post hoc confirmation of species identity. They were stored in 1.2 ml Eppendorf tubes with 99.8% ethanol for subsequent DNA isolation. The research was approved by the Ethics Committee of Masaryk University. The approval number which allows us to work with vertebrate animals is CZ01308.

Figure 1 Sampling localities in Lake Tanganyika with indication of host species (photos by Wolfgang Gessl).

Morphometrics

The morphometric characterisation was based on 26 different metrics measured according to Řehulková, Mendlová & Šimková (2013) and Gillardin et al. (2012). Measurements and photos were taken using the same configuration as above. In some cases an extra magnification of 2× had to be used. Voucher specimens were deposited in the invertebrate collection of the Royal Museum for Central Africa (Tervuren, Belgium) under accession numbers MRAC 37792-802.

DNA extraction and genetic characterisation

Ethanol was removed by evaporation in a vacuum centrifuge. DNA was extracted using the Qiagen Blood and Tissue Isolation Kit according to the manufacturer’s instructions with some modifications (samples in ATL buffer (180 µl) with protein kinase (20 µl) were kept in 1.5 ml Eppendorf tubes overnight at room temperature). The DNA extract was then concentrated to a volume of 80 µl in 1.5 ml Eppendorf tubes using a vacuum centrifuge and stored at a temperature of −20°C until polymerase chain reaction amplification. Part of the 18S nuclear ribosomal DNA gene, together with the first Internal Transcribed Spacer (ITS-1) region was amplified for 5 individuals using the S1 (5′-ATTCCGATAACGAACGAGACT-3′) (Sinnappah et al., 2001) and IR8 (5′-GCAGCTGCGTTCTTCATCGA-3′) (Šimková et al., 2003) primers. Each amplification reaction contained 1.5 unit of Taq Polymerase, 1X buffer containing 0.1 mg/ml BSA, 1.5 mM MgCl2, 200 mM dNTPs, 0.5 mM of each primer and 30 ng of genomic DNA in a total reaction volume of 30 µl under the following conditions: 2 min at 94°C, 39 cycles of 1 min at 94°C, 1 min at 53°C and 1 min and 30 s at 72°C, and finally 10 min at 72°C . The obtained nucleic acid sequences were aligned using MUSCLE (Edgar, 2004) under default distance measures and sequence weighting schemes as implemented in MEGA 6.06 (Tamura et al., 2013), together with previously published sequences of Cichlidogyrus from ‘G.’ pfefferi (GenBank accession numbers KT037169, KT037170, KT037171, KT037172, KT037173; Vanhove et al., 2015). Sequences and their alignment were visually inspected and corrected using the same software. Uncorrected pairwise distances were calculated in MEGA. The newly obtained haplotype sequence was deposited in NCBI GenBank under accession number KT692939.

Results

All 73 adult monogeneans collected from ‘G.’ pfefferi specimens were identified as C. irenae following the original description of Gillardin et al. (2012). The prevalence was 83.3%, mean infection intensity 18.2 and mean abundance 15.1 (calculated using adult monogeneans only). Although there are slight differences visible, mainly in the dorsal anchors and the attachment of the accessory piece to the base of the copulatory tube, our set of measurements matches with the original description of C. irenae (Gillardin et al., 2012) (Table 1). Differences in heel length are caused by different metrics (measuring up to the base of the heel versus to the base of the copulatory tube).

Table 1 Comparison of measurements (in μm) on Burundese Cichlidogyrus irenae with the original description.

	C. irenae from Burundi (n = 30a)	C. irenae (Gillardin et al., 2012)	
Ventral anchor			
Total length	30.3 ± 2.3b (n = 28)c; (26.9–36.4)d	31.4 ± 1,6 (n = 14); (29.3–34.6)	
Length to notch	25.7 ± 0.9 (n = 25); (22.6–29.8)	28.5 ± 1.4 (n = 14); (26.1–30.2)	
Inner root length	8.7 ± 1.7 (n = 24); (5.6–10.8)	8.1 ± 1.3 (n = 14); (5.9–10.1)	
Outer root length	5.5 ± 0.7 (n = 18); (4.9–6.8)	5.4 ± 1.2 (n = 14); (3.2–7.8)	
Point length	8.5 ± 1.1 (n = 25); (6.9–10.4)	10.0 ± 1.5 (n = 14); (7.9–12.8)	
Dorsal anchor			
Total length	30.5 ± 2.6 (n = 22); (27–37.5)	35.0 ± 2.8 (n = 15); (30.0–38.5)	
Length to notch	21.8 ± 1.1 (n = 16); (19.8–23.9)	25.8 ± 1.6 (n = 15); (22.4–28.8)	
Inner root length	10.6 ± 1.3 (n = 16); (7.9–13.4)	12.3 ± 1.5 (n = 15); (9.6–14.7)	
Outer root length	5.3 ± 0.9 (n = 16); (4,1–7,2)	4.6 ± 0.7 (n = 15); (3.6–5.9)	
Point length	7.1 ± 1 (n = 12); (5.7–8.7)	9.1 ± 1.0 (n = 15); (6.9–11.1)	
Ventral bar			
Branch length	38.4 ± 4.4 (n = 22); (32–49.5)	31.6 ± 4.6 (n = 15); (24.8–39.5)	
Branch maximum width	6 ± 0.9 (n = 28); (3.6–8.1)	4.8 ± 0.9 (n = 15); (3.2–6.5)	
Dorsal bar			
Maximum straight width	40.1 ± 4.1 (n = 14); (35–48.6)	32.7 ± 7.0 (n = 15); (17.9–45.8)	
Thickness at middle length	7.5 ± 1.2 (n = 28); (5.7–10.3)	6.1 ± 1.1 (n = 15); (4.2–8.2)	
Distance between auricles	15.2 ± 1.9 (n = 28); (12.1–18.4)	11.5 ± 1.8 (n = 15); (8.3–15.2)	
Auricle length	15.3 ± 2.3 (n = 15); (12.2–19.9)	14.2 ± 2.4 (n = 15); (9.6–19.0)	
Hooks			
Pair I	12.3 ± 0.6 (n = 26); (11.5–13.2)	11.6 ± 0.4 (n = 15); (10.8–12.1)	
Pair II	18.5 ± 2.1 (n = 28); (14.8–22.8)	–	
Pair II	20.6 ± 1.2 (n = 25); (18.4–22.2)	–	
Pair IV	21.1 ± 1.5 (n = 25); (19.4–25)	–	
Pair V	10.1 ± 0.9 (n = 10); (9.4–12.2)	11.4 ± 0.9 (n = 15); (9.2–12.6)	
Pair VI	21.4 ± 2.4 (n = 10); (16.1–22.8)	–	
Pair VII	20.6 ± 3.3 (n = 18); (17.5–25.7)	–	
Average size of pairs II, III, IV, VI, VII	20.2 ± 2.5 (n = 105); (13.3–27.3)	16.3 ± 2.1 (n = 15); (11.9–19.3)	
Copulatory tube curved length	69.9 ± 5.3 (n = 30); (59.3–81.4)	69.5 ± 5.7 (n = 20); (48.0–73.3)	
Accessory piece curved length	68.8 ± 8.2 (n = 30); (54–91)	59.5 ± 5.8 (n = 20); (37.8–64.8)	
Heel straight length	11.1 ± 3.9 (n = 30); (6–12.6)	4.1 ± 0.2 (n = 20); (3.6–4.4)	
Notes.

a Number of specimens.

b Standard deviation.

c Number of specimens.

d Range.

Only one specimen of G. permaxillaris was infected by monogeneans. It carried a single representative of a species of Cichlidogyrus similar in morphology to C. gillardinae parasitizing on Astatotilapia burtoni. Unfortunately, we cannot confidently confirm conspecificity based on only one specimen and therefore we refer to it as C. cf. gillardinae. Its pairs of anchors are asymmetrical: the dorsal anchor has a much longer guard than shaft while in the ventral anchor, guard and shaft are equal in size. The auricles and ventral bar branches are relatively short. Its male copulatory organ is characterised by a short heel, a simple copulatory tube with constant diameter and an accessory piece with easily overlooked distal bulb. No sclerotized vagina was observed. Despite these similarities with C. gillardinae, some differences compared to the original description were noted, e.g., Cichlidogyrus cf. gillardinae from G. permaxillaris has a more slender heel and shorter ventral anchor roots (Table 2).

Table 2 Comparison of measurements (in μm) on Burundese Cichlidogyrus cf. gillardinae with the original description.

	C. cf. gillardinae from Burundi (n = 1)a	C. gillardinae (Muterezi Bukinga et al., 2012) (n = 30)a	
Ventral anchor			
Total length	29.5	32 (27–37)	
Length to notch	26	28 (23–32)	
Inner root length	6.5	10 (8–13)	
Outer root length	3.8	6 (4–9)	
Point length	10.8	8 (6–11)	
Dorsal anchor			
Total length	31	33 (29–38)	
Length to notch	22.5	23 (19–29)	
Inner root length	10.5	12 (9–16)	
Outer root length	4.6	5 (4–7)	
Point length	7.75	7 (5–8)	
Ventral bar			
Branch length	29	31 (27–35)	
Branch maximum width	3.7	5 (3–6)	
Dorsal bar			
Maximum straight width	33	33 (27–39)	
Thickness at middle length	6.5	6 (4–8)	
Distance between auricles	11.8	12 (9–15)	
Auricle length	9.3	11 (8–14)	
Hooks			
Pair I	14.5	11 (9–13)	
Pair II	13.5	14 (11–17)	
Pair III	15.1	21 (18–26)	
Pair IV	21.5	22 (19–24)	
Pair V	9.5	10 (8–12)	
Pair VI	21.5	15 (13–17)	
Pair VII	14.1	17 (15–21)	
Copulatory tube curved length	51	47 (42–55)	
Accessory piece curved length	30	35 (29–42)	
Heel straight length	6.5	5 (4–7)	
Notes.

a Number of specimens.

Two monogenean specimens of an undescribed species of Cichlidogyrus were collected from one individual of L. auritus. One of the most noticeable structures within this parasite’s haptor are the extremely long auricles of the dorsal transverse bar. There is no visible difference between the length of guard and shaft in any of the anchors. The copulatory tube is thin with a constant diameter; a heel was not recognized. The accessory piece is robust and thick with a fork-shaped ending. No sclerotized vagina was observed. In view of the remarkably long auricles, this species morphologically resembles C. vandekerkhovei and C. makasai MPM Vanhove, F Volckaert and A Pariselle, 2011 described from Opthalmotilapia J Pellegrin, 1904 species. However, there are clear differences in MCO structure. For example, the copulatory tube tapers distally in C. vandekerkhovei and C. makasai, whereas it is of constant diameter in the undescribed parasite of L. auritus.

Micrographs of the collected monogenean species are presented in Fig. 2.

Figure 2 Micrographs of haptoral and male genital sclerotized structures from monogenean species belonging to Cichlidogyrus.

Host species: (A) ‘G.’ pfefferi (opisthaptor, Hoyer’s medium, phasecontrast); (B) ‘G.’ pfefferi (MCO, Hoyer’s medium, phasecontrast); (C) G. permaxillaris (opisthaptor, GAP); (D) G. permaxillaris (MCO, GAP); (E) L. auritus (opisthaptor, Hoyer’s medium, phasecontrast); (F) L. auritus (MCO, Hoyer’s medium, phasecontrast).

The rDNA dataset included four successfully amplified sequences of parasites collected from ‘G.’ pfefferi. Only one haplotype (1,060 base pairs) was recognised. The maximum overlap with sequences of more southern parasites of ‘G.’ pfefferi obtained from GenBank was 571 base pairs, situated within ITS-1. The uncorrected pairwise genetic distance reached a maximum of 0.8%, which is below the species-level cut-off of 1%, suggested for this region for the best-studied monogenean, Gyrodactylus A von Nordmann, 1832 (Ziętara & Lumme, 2002). This result confirms the identification, based on morphology and morphometrics, of a single monogenean species infecting ‘G.’ pfefferi, namely C. irenae.

Figure 3 Geographical position of records of C. irenae, monogeneans infecting ‘G.’ pfefferi.

Discussion

The monogenean fauna of the cichlid ‘G.’ pfefferi in Burundi was characterised morphologically and genetically. We confirmed the occurrence of C. irenae, representing the first record of this species in Burundi. According to previous results, the species richness of Cichlidogyrus on Tanganyika cichlids is influenced by the dispersal ability or isolation of the host species (Pariselle et al., 2015a; Grégoir et al., 2015). Although some differences in the size of parasite sclerotized structures were recorded (Table 1), these are only minor and likely reflect phenotypic intraspecific variability across entire Lake Tanganyika. Our results therefore support previous suggestions that ‘G.’ pfefferi, as a cichlid with good dispersal ability, hosts only a single representative of Cichlidogyrus, now recorded from several localities in the northern as well as the southern part of the Lake (Vanhove et al., 2015) (see Fig. 3).

Monogenean parasites belonging to Cichlidogyrus were also used as an additional way to look at species interrelationships within ‘Gnathochromis.’ The parasite from G. permaxillaris was identified as C. cf. gillardinae. Since C. gillardinae was originally described from the haplochromine A. burtoni, a fish occurring in aquatic systems along Lake Tanganyika’s shores, it is most likely a generalist parasite infecting representatives of two unrelated cichlid genera with different habitat preferences (Konings, 1998; Muterezi Bukinga et al., 2012). Although the limnochromine G. permaxillaris is hence infected by a monogenean species different from C. irenae described from ‘G.’ pfefferi, its parasite seems more similar to its congeners infecting tropheine hosts like ‘G.’ pfefferi (Gillardin et al., 2012; Pariselle et al., 2015b). Cichlidogyrus can be divided into different lineages based on the configuration of their haptoral hard parts, in particular the relative length of the pairs of hooks (also termed uncinuli) (Pariselle & Euzet, 2003; Vignon, Pariselle & Vanhove, 2011). Indeed, both parasites’ haptor shares important characteristics: asymmetry between anchors, small (sensuPariselle & Euzet, 2009) hooks. Cichlidogyrus cf. gillardinae differs substantially from the Cichlidogyrus species collected from the closely related host L. auritus, another limnochromine cichlid. In the latter flatworm, the extremely long dorsal bar auricles represent an evident similarity with C. vandekerkhovei and C. makasai (Vanhove, Volckaert & Pariselle, 2011) collected from species of Ophthalmotilapia, belonging to the Ectodini, another cichlid tribe endemic to Lake Tanganyika. This feature was hitherto never found in other monogenean congeners. The gill monogenean retrieved from Limnochromis hence seems to belong to an endemic Tanganyika lineage. The discussion about the evolution of the haptoral sclerotized structures is still ongoing. Morand et al. (2002) assume that haptoral structures do not reflect a phylogenetic pattern as a result of adaptation to microhabitat within the host. Moreover, Messu Mandeng et al. (2015) point out an adaptive component in the attachment organ morphology of Cichlidogyrus. However, other studies suggest the existence of a phylogenetic signal in sclerite morphology and shape within dactylogyridean monogeneans (Šimková et al., 2002; Šimková et al., 2006) and specifically within Cichlidogyrus (Vignon, Pariselle & Vanhove, 2011).

According to Mendlová & Šimková (2014) the host specificity of Cichlidogyrus parasitising African cichlid fishes is significantly influenced by fish phylogeny and by form of parental care. No Cichlidogyrus species was hitherto observed to infect cichlid species with different parental care systems (i.e., substrate brooders as well as mouthbrooders) (Pouyaud et al., 2006). However, the form of parental care in cichlids is directly influenced by phylogenetic history and relationships (Goodwin, Balshine-Earn & Reynolds, 1998). Possible explanations for the affinities of monogenean species on ‘Gnathochromis’ are therefore host evolutionary history as well as habitat characteristics. While ‘G.’ pfefferi is a typical rock dwelling littoral cichlid occurring at depths between 1 and 15 m, G. permaxillaris occurs over muddy bottoms and is rarely seen in water shallower than 30 m (Maréchal & Poll, 1991; Konings, 1998). Limnochromis auritus is placed together with G. permaxillaris in the Limnochromini and prefers similar habitats with muddy bottoms at depths ranging from 5 to 125 m (Maréchal & Poll, 1991; Konings, 1998). Given that the haplochromine A. burtoni occurs in wetlands adjacent to the lake, in river mouths and in vegetated areas in the lake proper, it is unclear how it came to share a species with G. permaxillaris from which it differs ecologically and phylogenetically. On the other hand, the deepwater limnochromines G. permaxillaris and L. auritus seem to host entirely different monogeneans. However, these findings are based on a limited number of specimens (only one specimen of Cichlidogyrus collected from G. permaxillaris). Due to the lack of genetic data, we cannot perform (co-)phylogenetic analyses. According to Mendlová et al. (2012) duplication and host-switching events have played the most important role in the evolutionary history of African cichlid dactylogyridean species. Vanhove et al. (2015), however, found evidence for an important role of co-speciation in the evolution of Cichlidogyrus infecting Lake Tanganyika’s tropheine cichlids. Although representatives of Cichlidogyrus occuring on littoral cichlid assemblages including Tropheini display strong host specificity (Gillardin et al., 2012; Muterezi Bukinga et al., 2012; Vanhove et al., 2015), a lower specificity was observed within the Bathybatini, a deepwater cichlid tribe from Lake Tanganyika (Pariselle et al., 2015a). Hence, some lineages of Cichlidogyrus in Lake Tanganyika were already shown to have a wide host range. The observed low host specificity and the apparent low infestation rate most likely correlate with low host density in the deepwater habitat (Justine et al., 2012; Schoelinck, Cruaud & Justine, 2012). Given the low prevalence and infection intensities observed in this study, and the deepwater habitat of the limnochromine hosts, it is a challenge to retrieve additional material for species identification and molecular analyses. These, together with a broadened geographical coverage, are needed to uncover the whole co-phylogenetic history of ‘Gnathochromis’ and its monogenean fauna.

Supplemental Information

Supplemental Information 1 Measurements of Cihlidogyrus irenae

Click here for additional data file.

We would like to thank Maarten Van Steenberge, Tine Huyse and the parasitology group at Masaryk University, Brno for their hospitality and cooperation. Eva Řehulková, Šárka Mašová, Iva Přikrylová, Radim Blažek, Veronika Nezhybová, Gaspard Banyankimbona and the Schreyen-Brichard family and the technical staff of Fishes of Burundi are thanked for their help in collecting samples, and Wolfgang Gessl for providing fish pictures. Antoine Pariselle, Miguel Rubio-Godoy and an anonymous referee are acknowledged for their constructive comments.

Additional Information and Declarations

Competing Interests

Author Contributions

Animal Ethics

DNA Deposition

Data Availability

The authors declare there are no competing interests.

Nikol Kmentová conceived and designed the experiments, performed the experiments, analyzed the data, wrote the paper, prepared figures and/or tables.

Milan Gelnar contributed reagents/materials/analysis tools, reviewed drafts of the paper.

Stephan Koblmüller analyzed the data, reviewed drafts of the paper.

Maarten P.M. Vanhove conceived and designed the experiments, analyzed the data, wrote the paper.

The following information was supplied relating to ethical approvals (i.e., approving body and any reference numbers):

The research was approved by Ethics Committee of Masaryk University, approval number CZ01308.

The following information was supplied regarding the deposition of DNA sequences:

GenBank accession number KT692939.

The following information was supplied regarding data availability:

The research in this article did not generate any raw data.

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
