# Peer review of "First insights into the diversity of gill monogeneans of ‘Gnathochromis’ and Limnochromis (Teleostei, Cichlidae) in Burundi: do the parasites mirror host ecology and phylogenetic history?"

_PeerJ, doi:10.7717/peerj.1629_

## Round 0.1 · original submission · Minor Revisions

Please consider all the suggestions in your revised manuscript, and please note the 2 documents supplied by Reviewers 2 and 3.

·

Basic reporting

No Comments

Experimental design

No Comments

Validity of the findings

No Comments

Additional comments

This paper presents concise, elegant data demonstrating the utility of monogenean parasites to shed light on the probable evolutionary history of members of the extremely diverse cichlid fish fauna of the Great African Lakes. Moreover, it demonstrates that within the great diversity of fish hosts, probably lies an unexplored diversity of evolutionary and co-evolutionary trajectories, which the study of parasites may help illuminating. This is a nice contribution to the growing realization that parasites are an integral part of ecosystems, which cannot be fully understood without considering them; and also adds an interesting and as yet not fully exploited tool to accomplish integrative taxonomical descriptions of animal hosts: to include their parasite fauna in a holistic delimitation of a species.

·

Basic reporting

Photos of figure 2 have to be lightened

Experimental design

No comments

Validity of the findings

The study is interesting, it was well imagined (the use of parasites as host’s descriptor is increasing), designed (the choice of hosts and localities is pertinent) and leaded (field work is not easy in Africa), unfortunately authors could no sample enough monogenean specimens to provide pertinent analysis and sound conclusion.

Additional comments

In a “normal” journal I would have advised the editor against publishing this manuscript, but the standards of PeerJ allow the publication of such innovative manuscript (and it is a good think in my opinion), so I recommend the publication of this manuscript after minor revisions (see my remarks done directly into the text).

Reviewer 3 ·

Basic reporting

The paper is well written and conclusion drawn is valid. However, it is based on a small sample size and therefore I recommend rephrasing of some aspects. The taxon authors are not correctly cited/added and the authors should follow the taxonomic rules (unless these have changed and I am not aware of it).

Experimental design

The experimental design is good and well explained but as mentioned previously, based on a small sample size.

Validity of the findings

The findings are valid but some of the statements should be revised as conclusions were drawn based on a small sample size (and one parasite for one example mentioned). I have indicated these on the PDF with comments.

Additional comments

The paper is well written and give some insight into evolutionary history of parasites (monogeneans) and their hosts. However, these conclusions are based on a small sample size.

Annotated reviews are not available for download in order to protect the identity of reviewers who chose to remain anonymous.

---

## Round 0.2 · accepted · Accept

Thank you for the revision of your manuscript.